# Generalized Abs-Linear Learning

Andreas Griewank[0000−0001−9839−1473] and Ángel Rojas

Humboldt University, Berlin, and Yachay Tech, Imbabura, Ecuador

**Abstract.** We consider predictor functions $f(w; x)$ in abs-linear form, a generalization of neural nets with hinge activation. To train them with respect to a given data set of feature-label pairs $(x, y)$ one has to minimize the average loss, which is a multi-piecewise linear or quadratic function of the weights, i.e. coefficients of the abs-linear form. We suggest to attack this nonsmooth global optimization problem via successive piecewise linearization, which allows the application of coordinate search, gradient based methods or mixed binary linear optimization. These alternative methods solve the sequence of abs-linear model problems with a proximal term as demonstrated in [5]. More general predictor functions $f(w; x)$ that are given in abs-normal form can be successively abs-linearized with respect to the *weights* $w$, which can then be optimized in a nested iteration. In the talk we will present numerical validations and comparisons with standard methods like ADAM [11], e.g. on the MNIST problem.

## Introduction and Notation

Neural nets with hinge activation have been proven theoretically [13], [2], and experimentally [14] to produce prediction functions $f(w; x)$ that are able to represent a wide variety of relations in machine learning. These predictor models are piecewise linear [12] with respect to the feature vector $x$ and multi-piecewise-linearwith respect to the weight vector $w$, which consists of various transformation matrices and inhomogeneous shifts. It is well known that every piecewise linear function from $x \in \mathbb{R}^n$ to $y \in R^m$ can be expressed in an abs-linear form

$$y = f(w; x) \equiv Nz \quad s.t. \quad z = c + Zx + Mz + L|z| \tag{1}$$

where $z \in \mathbb{R}^s$ is a vector of *switching variables* and $M, L$ are strictly lower triangular. The coefficient vectors and matrices can be combined to the vector

$$w \equiv (c, Z, M, L, N) \in \mathbb{R}^{(s, s \times n, s \times s, s \times s, m \times s)} \simeq \mathbb{R}^{\bar{s}}$$

where $\bar{s} = s(s + m + s - 1)$, taking the strict lower triangularity into account. In the case of multi-layer neural networks $z$ represents nodal values and the matrices $M$ and $L$ are block diagonal, with $M = L$ in the case of hinge activation. More generally the matrices in $w$ can be restricted statically or adaptively during the learning process to achieve a certain sparsity pattern in order to reduce the spatial and temporal complexity.

---

[1] Corresponding author: Andreas Griewank, griewank@math.hu-berlin.de

By induction on the $s$ components $z_i$ of $z$ one can easily see that they are piecewise linear functions $z_i(x)$ of $x$, which then also holds for the resulting vector $y = f(w; x) = Nz(x) \in \mathbb{R}^m$. Our working hypothesis is that the abs-linear representation of piecewise linear functions is computationally more appropriate than the frequently used alternatives, in particular linear expansions in terms of hat functions. These are very spiky and do not provide a natural connection between the values of the prediction function at distinct sample points.

A typical empirical risk function may then look like

$$\varphi(w) \equiv \tfrac{1}{2d} \sum_{k=1}^{d} \|f(w, x_k) - y_k\|_2^2 \quad \text{with} \quad f(w; x) = Nz(w; x) \qquad (2)$$

where the (feature, label) pairs $(x_k, y_k) \in \mathbb{R}^{n \times m}$ for $k = 1, 2, \ldots d$ form a suitable training set. It is important to note that except in the single layer neural network case, the dependance of the predictor $f(w; x)$ on the coefficients $w$ is only *multi-piecewise linear*, i.e. piecewise linear with respect to each component of $w$ when the others are kept constant. In other words for each Cartesian basis vector $e_j \in \mathbb{R}^{\bar{s}}$ and fixed $x$ the univariate function $f(w + te_j; x)$ is piecewise linear. Therefore one can rather efficiently perform (global) coordinate searches (see Wright [10]), even when the loss function on $y$ is not piecewise linear but for example quadratic.

## Successive abs-linearization w.r.t. the weights

Notice that the general abs-linear form can approximate any continuous function with arbitrary accuracy, so it has no convexity properties whatsoever. However, it does allow one to deal with nonsmoothness explicitly. Also, via their reformulation in terms of Mixed Integer Linear Optimization problems (MILOP) [9], piecewise linear objectives can be globally minimized using modern branch and bound solvers like Gurobi. For training this approach can only be applied to local piecewise linearizations $\tilde{\varphi}(\mathring{w}; \Delta w)$ at a sequence of reference points $\mathring{w} = (\mathring{c}, \mathring{Z}, \mathring{M}, \mathring{L}, \mathring{N})$. It was shown in [4] that the piecewise linearization of the righthand side of (1) is given by the triangular system

$$\tilde{z} = (c + Zx + \Delta M \mathring{z} + \Delta L |\mathring{z}|) + \mathring{M} \tilde{z} + \mathring{L} |\tilde{z}| \ . \qquad (3)$$

where naturally $\Delta M = M - \mathring{M}$ and $\Delta L = L - \mathring{L}$. More specifically it was shown that when this approximating $\tilde{z} = \tilde{z}(\mathring{w}; \Delta w)$ is used to replace $z$ in (2) we obtain an approximate average loss $\tilde{\varphi}(\mathring{w}; \Delta w)$ such that for some constant $q$

$$|\varphi(w) - \tilde{\varphi}(\mathring{w}; \Delta w)| \leq \tfrac{q}{2} \|\Delta M, \Delta L\|_F^2 \leq \tfrac{q}{2} \|\Delta w\|_2^2$$

with $\|\cdot\|_F$ denoting the Frobenius norm of matrices. The changes $\Delta c, \Delta Z$ and $\Delta Z$ to the coefficients $\mathring{c}, \mathring{Z}$ and $\mathring{N}$ that enter linearly into (1) do not occur explicitly in the proximal term on the right hand side. In [4] we gave an explicit bound for the values of $q$, which tends to be rather large and thus too conservative for practical calculations.

The outer loop of the training problem now consists of generating iterates $w_k$ by the recursion

$$w_{k+1} \;=\; \underset{\Delta w}{\mathrm{argmin}}\{\tilde{\varphi}(w_k; \Delta w) + \tfrac{q}{2}\|\Delta w\|_2^2\} \;. \qquad (4)$$

For solving the inner loop problem on the right hand side we have considered besides coordinate search, the MILOP reformulation and a dynamic trajectory search method called TOAST of the class considered recently in [1]. For comparison purposes we also consider steepest descent and its moment variants as well as stochastic gradient. See also the contributions [3] and [8], where neural net models are considered as discretized ODEs or PDEs and optimized on training sets based on their continuous adjoints.

### Backpropagation w.r.t. weights

To minimize $\varphi(w)$ or its approximation $\tilde{\varphi}(\mathring{w}; w - \mathring{w})$ by the classical gradient based methods or our proposal TOAST one needs the derivatives with respect to all component vectors and matrices of $w$. By differentiation of the smooth $\ell_2$-loss (2) with respect to the shift $c \in \mathbb{R}^s$ one gets the adjoint vector

$$\bar{c} \;\equiv\; \nabla_c \varphi \;\equiv\; \frac{1}{d} \sum_{k=1}^{d} \bar{c}_k \quad \text{with} \quad \bar{c}_k = N^\top(f(w, x_k) - y_k) \;\in\; \mathbb{R}^s \;. \qquad (5)$$

Then one has to compute $\bar{z}_k$ as solution of the unit upper triangular linear system

$$(I - M^\top - \Sigma_k L^\top)\bar{z}_k = \bar{c}_k \in \mathbb{R}^s \quad \text{where} \quad \Sigma_k z_k \geqslant 0 \text{ and } \det(\Sigma_k) = \pm 1 \;.$$

This task proceeds naturally by backward substitution using exactly the same number of operations as computing $z_k$ from a given $x_k$. It can be shown that the adjoint values $\bar{w} = \nabla_w \varphi(w) = (\bar{c}, \bar{Z}, \overline{M}, \bar{L})$ are given by $\bar{c}$ as defined above and

$$\bar{Z} \;=\; \mathcal{S}_Z\left[\frac{1}{d}\sum_{k=1}^{d} \bar{z}_k x_k^\top\right], \quad \overline{M} \;=\; \mathcal{S}_M\left[\frac{1}{d}\sum_{k=1}^{d} \bar{z}_k z_k^\top\right], \quad \bar{L} \;=\; \mathcal{S}_L\left[\frac{1}{d}\sum_{k=1}^{d} \bar{z}_k |z_k|^\top\right] \;.$$

Here the linear operators $\mathcal{S}_Z, \mathcal{S}_L$ and $\mathcal{S}_Z$ may enforce any desired sparsity pattern of the matrices, including especially the strict lower triangularity of $M$ and $L$. Hence we see that the total effort for calculating $\varphi(w)$ and $\nabla_w \varphi(w)$ is almost exactly three times as much as evaluating $\varphi(w)$ as such, which requires one fused multiply add for each nonzero entry in $w$. This factor 3 agrees exactly with the multiplicative upper bound in the cheap gradient principle [6]. The derivative formulas above can be easily adopted to the piecewise linearized system (3) and the resulting objective $\tilde{\varphi}(\mathring{w}; w - \mathring{w})$.

At the time of writing we have only done some very preliminary calculations comparing the general abs-linear model to a one-layer neural net with the same number of degrees of freedom in the coefficients. The learning task was regression on the Griewank function [7] with feature dimension 4 and a training set of 20 data points. The same problem was used in Fig. 1 to compare four methods on the training of a neural net with a single intermediate layer of 10 nodes and inhomogeneous shifts. For a detailed description of TOAST see [5] and the talk.

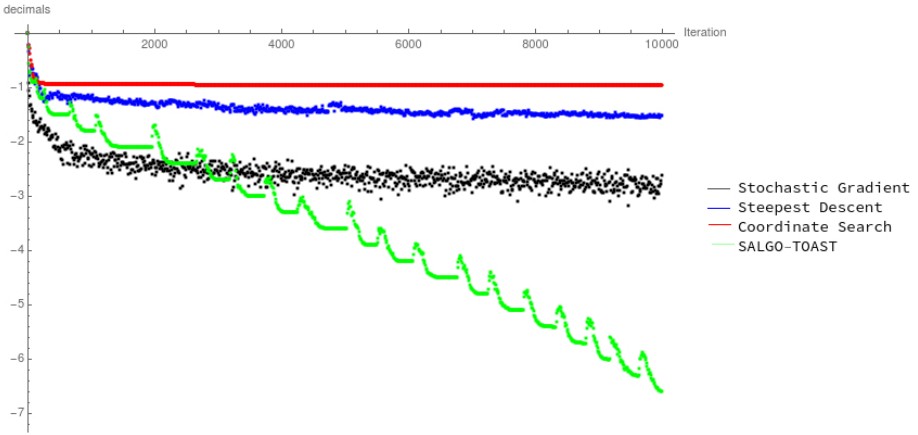

**Fig. 1.** Decimal digits gained by four methods on single layer regression problem.

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
