# OpenReview forum: "Generalized Abs-Linear Learning"
_NeurIPS.cc/2019/Workshop/Program_Transformations — Program Transformations @NeurIPS2019 Oral_

### Official Review · AnonReviewer2 · 2019-09-30
**Takes hinge nondifferentiability seriously**

**Confidence:** 4
**Rating:** 9

**Review:**

It would be hilarious if methods that justify hinge functions (RELU nonlinearities), and more sophisticated/better optimization methods than simple gradient methods applied (as training proceeds) precisely at points where the gradient is not well defined, came from the traditional AD community. This approach takes the problem seriously, unlike the entire deep learning community. In particular, mathematical methods for really handling abs(x) in the primal computation, are I think of interest. Not just from the perspective of building networks, but also as something implementors of AD systems targeted for machine learning should know about, and might consider supporting.

I can imagine an oral about this work giving a good intuition for how these issues of discontinuities in piecewise-linear models can be handled in a principled way, and how much computational overhead this entails.

---

### Official Review · AnonReviewer1 · 2019-09-30
**Interesting approach, using knowledge of the piecewise linear landscape to optimize neural nets**

**Confidence:** 2
**Rating:** 6

**Review:**

The article proposes a way to exploit the piecewise-linear structure of the function represented by a neural network (with ReLU or other piecewise linear activation) by sophisticated optimization methods.
This is an interesting idea, well formalized, and the preliminary results on a toy problem are promising.
Unfortunately, the evaluation is still preliminary, and reference [5] does not seem to be available yet.
If the paper is selected as an oral, I would like to see the impact of using a better optimization method on the generalization performance. I'm also curious to see if a stochastic version is viable.

---

### Decision · Program_Chairs · 2019-10-01

**Decision:**

Accept (Oral)

**Comment:**

The reviewers believed this was a strong contribution that brings mathematical methods from an AD background into ML applications. The reviewers had questions though about the applicability of this method in practical deep learning (e.g., minibatches, generalization) that we would like to see discussed in the oral presentation.